# Glider Observations of Thermohaline Staircases in the Tropical North Atlantic Using an Automated Classifier

Callum Rollo[1], Karen J. Heywood[1], and Rob A. Hall[1]

[1]Centre for Ocean and Atmospheric Sciences, School of Environmental Sciences, University of East Anglia, Norwich, United Kingdom

**Correspondence:** Callum Rollo (callum.rollo@voiceoftheocean.org)

**Abstract.**

Thermohaline staircases are stepped structures of alternating thick mixed layers and thin high gradient interfaces. These structures can be up to several tens of metres thick and are associated with double-diffusive mixing. Thermohaline staircases occur across broad swathes of the Arctic and tropical/subtropical oceans and can increase rates of diapycnal mixing by up to five times the background rate, driving substantial nutrient fluxes to the upper ocean. In this study, we present an improved classification algorithm to detect thermohaline staircases in ocean glider profiles. We use a dataset of 1162 glider profiles from the tropical North Atlantic collected in early 2020 at the edge of a known thermohaline staircase region. The algorithm identifies thermohaline staircases in $97.7\,\%$ of profiles that extend deeper than 300 m. We validate our algorithm against previous results obtained from algorithmic classification of Argo float profiles. Using fine resolution temperature data from a fast-response thermistor on one of the gliders, we explore the effect of varying vertical bin sizes on detected thermohaline staircases. Our algorithm builds on previous work by adding improved flexibility and the ability to classify staircases from profiles with noisy salinity data. Using our results, we propose that the incidence of thermohaline staircases is limited by strong background vertical gradients in conservative temperature and absolute salinity.

## 1 Introduction

Thermohaline staircases consist of subsurface layers of near homogeneous salinity and temperature (referred to here as mixed layers), separated by thin layers with large temperature and salinity gradients (interfaces). A mixed layer separated by two interfaces is referred to as a step. Staircases formation and growth are driven by double-diffusive processes that arise from the difference in molecular diffusivities of heat and salt. Heat diffuses 100 times faster than salt (Stern, 1960). Thermohaline staircases form where the vertical gradients of temperature and salinity have the same sign. These conditions most commonly occur in the Arctic, where cool fresh waters overlie warm salty waters, and tropical/subtropical regions where warm salty surface waters overlie cool, fresh waters.

Thermal expansion and haline contraction are critical to the formation of thermohaline staircases. Their are often represented by either of two derived properties, the density ratio $R_\rho = \alpha\theta_z/\beta S_z$ and the Turner angle $Tu = tan^{-1}(\alpha\theta_z - \beta S_z, \alpha\theta_z + \beta S_z)$, where $\alpha$ is the thermal expansion coefficient, $\beta$ is the haline contraction coefficient, $\theta_z$ and $S_z$ are the vertical gradients in con-

servative temperature and absolute salinity (Ruddick, 1983) . The density ratio can be calculated from the Turner angle using the relationship $R_\rho = -tan(Tu + 45°)$ (You, 2002). The Turner angle categorises the water column into portions that are either statically unstable, prone to salt fingers, prone to diffusion-convection or doubly stable (Ruddick, 1983). Fig. 1 shows vertical profiles of conservative temperature and absolute salinity from an example thermohaline staircase from the Mediterranean (a&b). The figure also shows how Turner angle and density ratio relate (c).

In this study of the tropical North Atlantic, warm salty Subtropical Underwater (conservative temperature 22 - 25 °C, absolute salinity 37.0 - 37.6 g kg$^{-1}$) overlies cooler fresher Antarctic Intermediate Water (4-6 °C, 34.7-35.0 g kg$^{-1}$) (Fer et al., 2010). For the water masses in this location, the mechanism driving mixing in the interfaces is salt finger instability (Radko, 2005). This instability can spontaneously transform smooth, statically stable profiles into stepped patterns such as those in Fig. 1. This instability can occur where $45° \leq T_u \leq 90°$ (Fig. 1.c). Laboratory and theoretical studies have concluded that diapycnal mixing rates of heat and salt are elevated in thermohaline staircases (Schmitt, 1981 & Radko and Smith, 2011). Schmitt et al. (2005) estimated that thermohaline staircases raise diapycnal mixing rates in the tropical North Atlantic by a factor of five. Mixing rates control critical oceanographic processes including nutrient fluxes (Oschlies et al., 2003) and the meridional overturning circulation (Kuhlbrodt et al., 2007). Recent studies estimate that the contribution of thermohaline staircases to the global ocean mechanical energy budget is small, however, the regional contribution to mixing in locations of strong staircasing can be pronounced (van der Boog et al., 2021a). Due to their effect on mixing, the global incidence of thermohaline staircases, and the mechanisms that govern them, are the subject of ongoing research. This study aims to improve detection and quantification of thermohaline staircases using ocean gliders.

The tropical North Atlantic has been known to host thermohaline staircases since the 1970s (Schmitt et al. (1987), and references therein). Using ship CTD casts and expendable bathythermographs (XBTs) from ships and aircraft, thermohaline staircases were identified over an area exceeding one million square kilometres by the C-SALT experiment (Schmitt et al., 1987). Staircases were observed at depths of 420-650 m, with individual mixed layers 5-30 m thick. Subsequent studies using a fast-response thermistor chain (Marmorino et al., 1987), tracer release and fine-structure high resolution profilers (Schmitt et al., 2005), seismic oceanography (Fer et al., 2010) and Argo floats (van der Boog et al., 2021b, henceforth VDB), have observed staircases with similar bulk statistics in the tropical North Atlantic. See Table 5 of VDB for more details. The areas identified as strong or weak staircasing by Schmitt et al. (1987) appear to be consistent in time, with subsequent studies by Marmorino et al. (1987) and Fer et al. (2010) observing breakdowns of staircases at the edges of these areas.

While the exact formation mechanisms of thermohaline staircases are still a topic of discussion (Radko, 2020, VDB), several key conditions have been identified. The density ratio must be within a critical range. The lower bound for $R_\rho$ is 1, where the vertical gradients in salinity and temperature balance and there is no density stratification. Bryden et al. (2014), Schmitt et al. (1987) and Fer et al. (2010) all observed well ordered staircases up to $R_\rho = 1.7$, with irregular "steppy" profiles up to $R_\rho$ of 2.0 or greater. Theoretical work by Radko (2005) suggests that staircases can only form at $R_\rho < 2$. Fer et al. (2010) consider the range $1 < R_\rho < 2.5$ as conducive to forming thermohaline staircases in the tropical North Atlantic. Energetic features such as velocity shear, slope currents and internal waves can prevent the formation of staircases (Fer et al., 2010; Buffett et al., 2017); though in some cases, the staircases are strong enough to resist breakup (Schmitt et al., 1987), and can even dampen internal

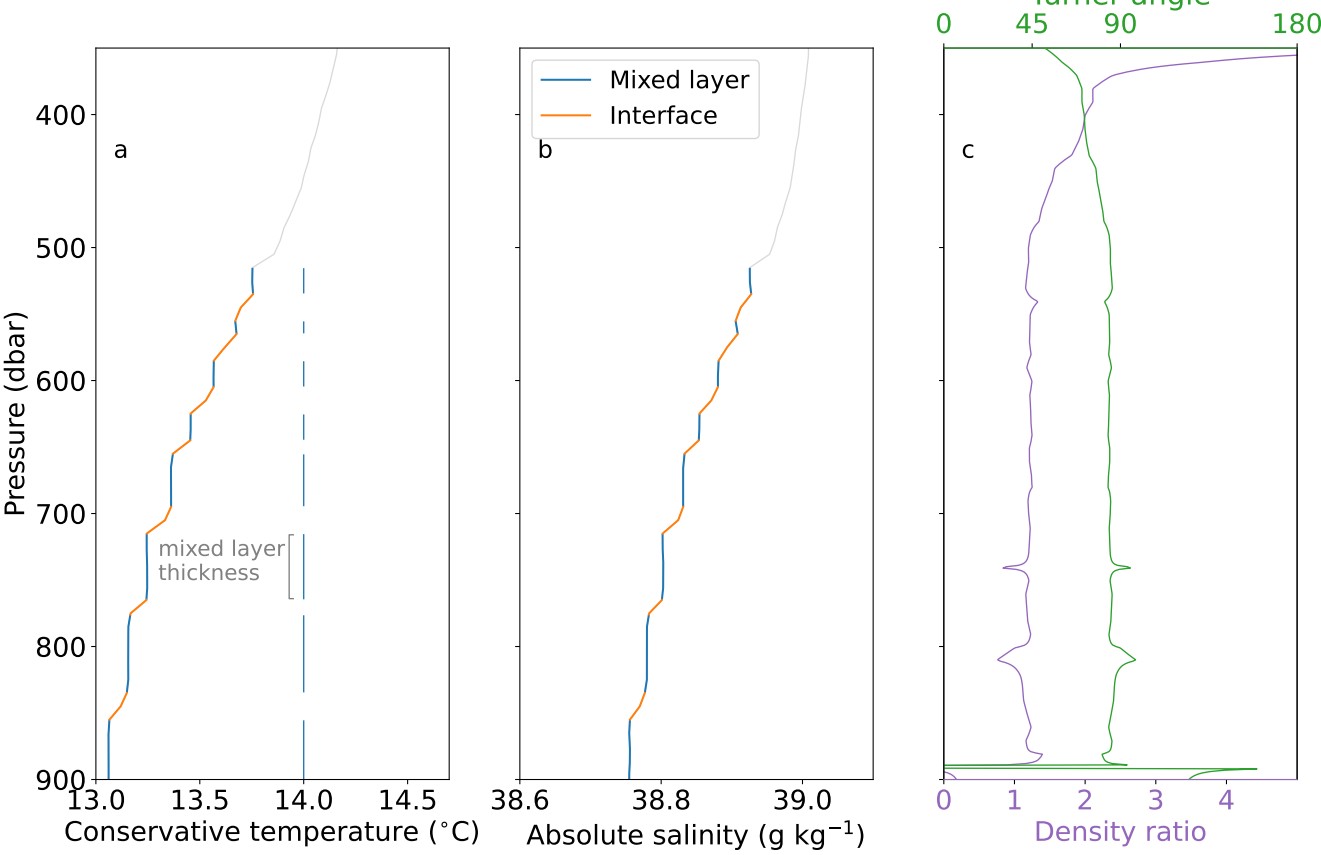

**Figure 1.** Example thermohaline staircase from Argo float 6901769 at 8.9° E, 37.9° N in the Mediterranean. Grey line is 1 dbar averaged temperature (a) and salinity (b), with blue and orange lines marking the mixed layers and interfaces of the staircase. (c) Corresponding profiles of Turner angle and density ratio.

wave activity (Radko, 2020). In this paper we suggest that strong vertical gradients in salinity and temperature also constrain the formation of thermohaline staircases.

To explore the incidence and structure of thermohaline staircases in the tropical North Atlantic, we analysed over 1000 profiles collected by three ocean gliders in early 2020. Gliders have previously been used to observe thermohaline staircases in the Tyrrhenian Sea in the Mediterranean, primarily to relate seismic reflectors to thermohaline structures (Buffett et al., 2017).

However this study is, to our knowledge, the first of its kind in the tropical North Atlantic. This is also the first study to identify thermohaline staircases in glider profiles algorithmically. Data collection is detailed in Sect. 2. The thermohaline classifier is described in Sect. 2.2. Results are presented in Sect. 3.1 and discussed in Sect. 3.3, with subsections on scale sensitivity and classifier limitations in Sect. 3.5 and Sect. 3.2. We summarise our conclusions and recommendations for future avenues of study in Sect. 4.

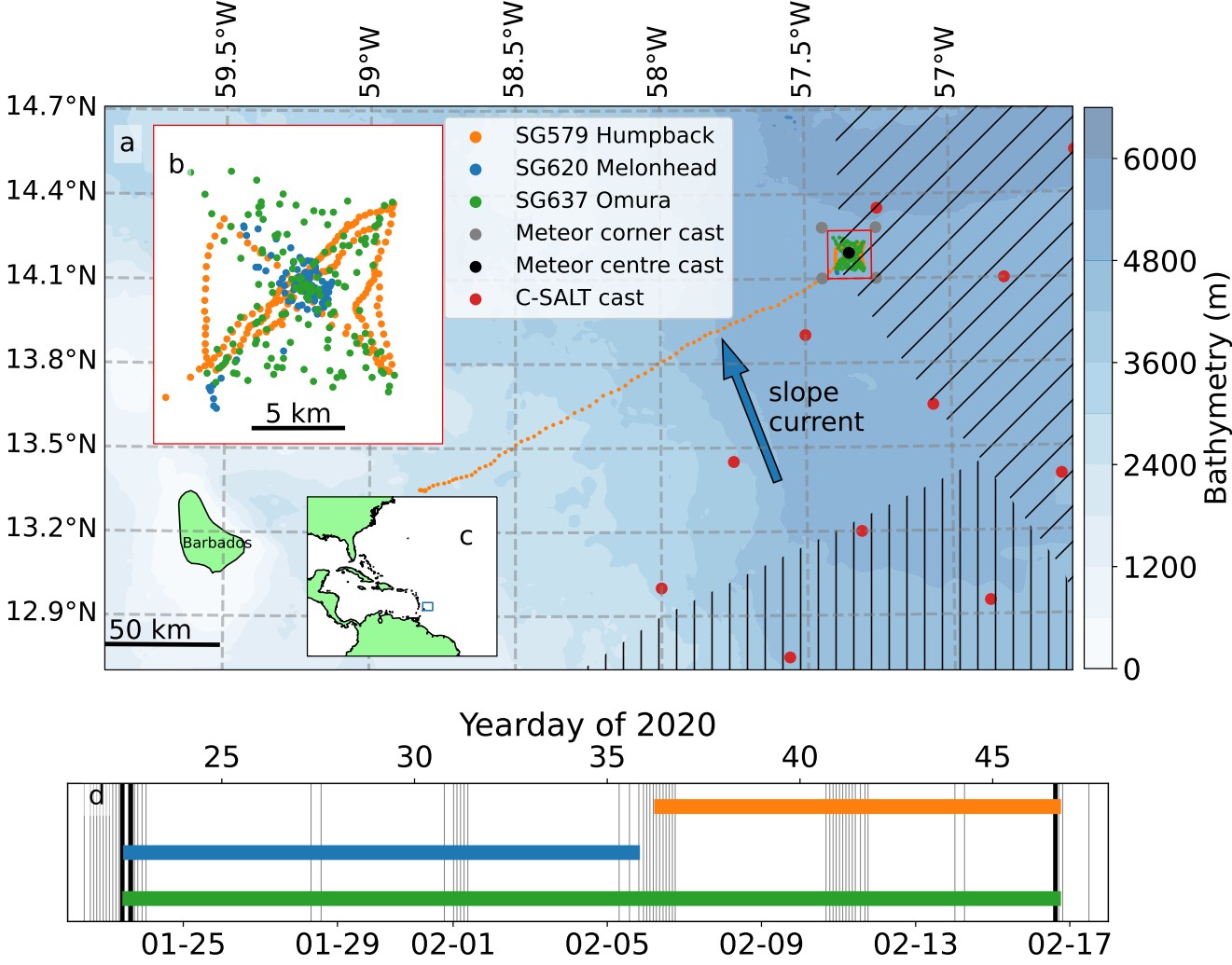

**Figure 2.** (a) Glider survey map showing glider and CTD data from the Tropical North Atlantic. Red dots are CTD casts conducted during the C-SALT experiment in Spring 1985 (Schmitt, 1987). Diagonal and vertical hatching respectively delineate the approximate areas of strong and weak staircases identified by Schmitt et al. (1987). (b) Zoomed detail of bowtie outlined in red on (a). (c) Map of the tropical North Atlantic, the blue box marks extent of (a). (d) Temporal overlap of glider observations at the bowtie and nearby *Meteor* CTD casts. *Meteor* corner casts were taken at grey dots on map. *Meteor* centre casts were taken at the black dot. Colours match legend in (a). The orange line for SG579 in this subplot covers only the period during which it was at the bowtie.

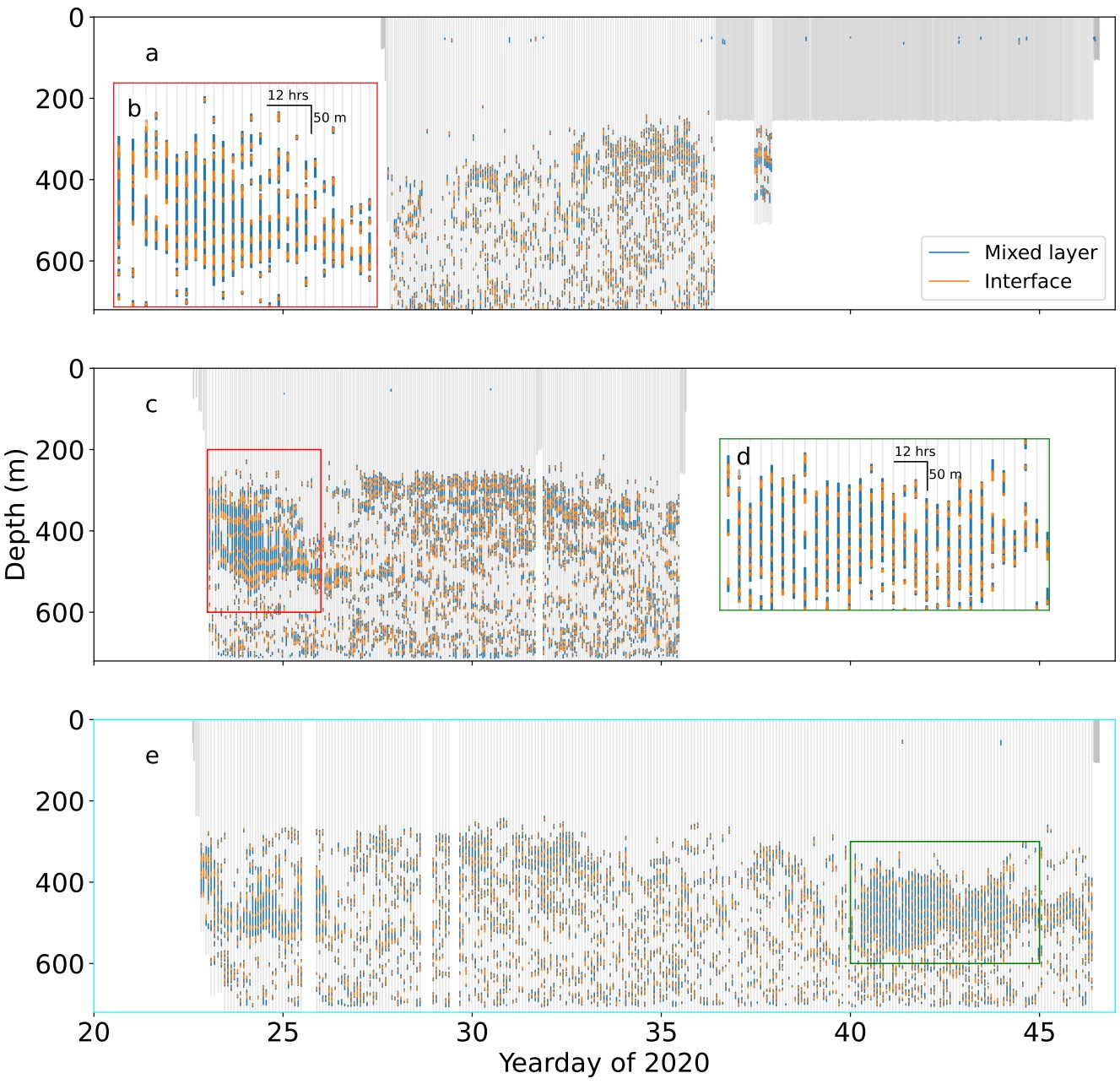

**Figure 3.** All thermohaline staircases detected in glider profiles. (a, c & e) data from gliders SG579, SG620, and SG637 respectively. Data from gliders in the Tropical North Atlantic. (b & d) show expanded sections from boxes in c & e in the corresponding colour. Data from (e) are the basis for remaining figures in this manuscript, except where otherwise stated.

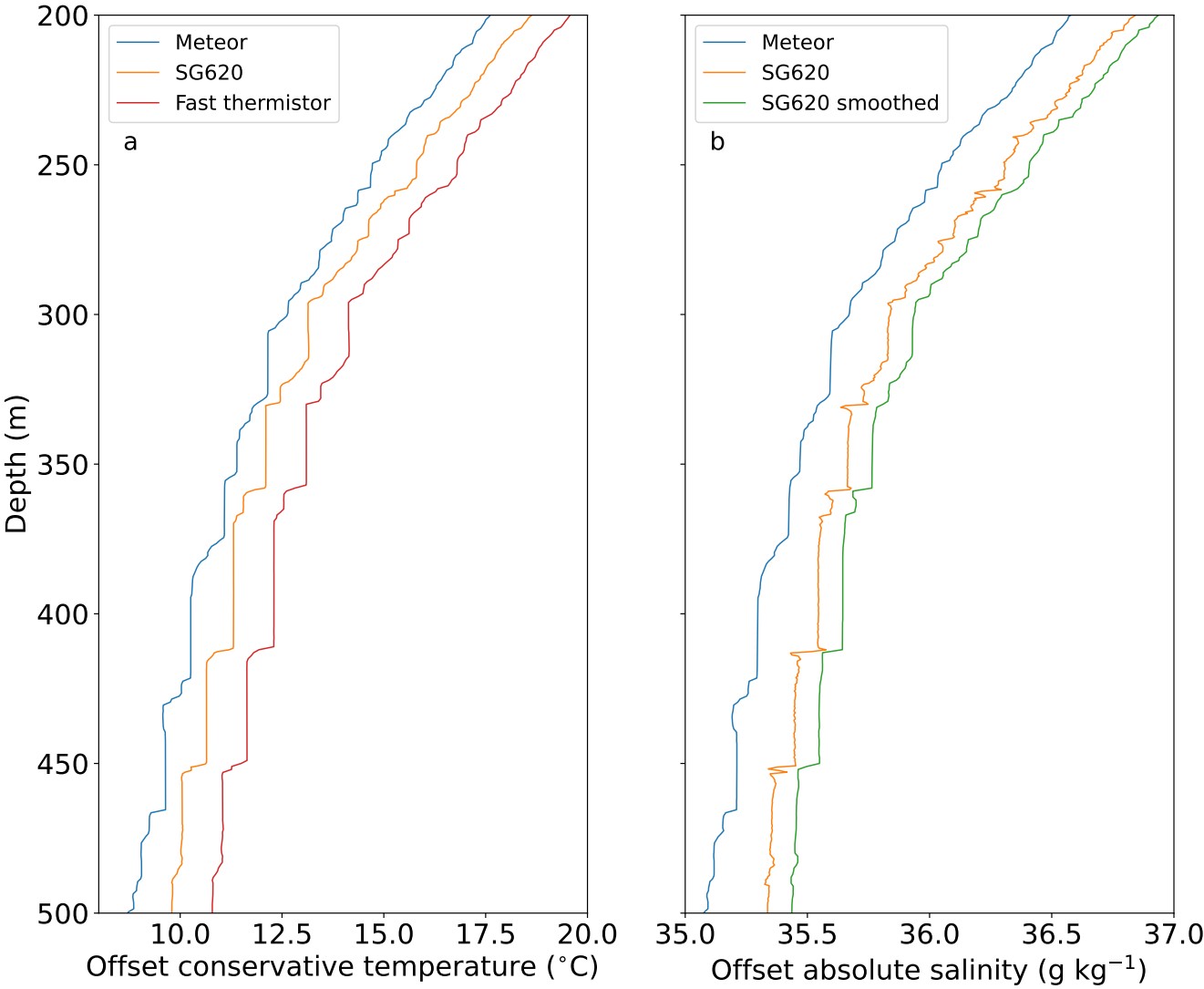

**Figure 4.** (a) Temperature profiles from *Meteor* CTD, SG620 CT sail and fast-response thermistor from the Tropical North Atlantic. (b) Salinity from *Meteor*, salinity from SG620 and SG620 salinity smoothed with a 5 m running mean. The profile from the *Meteor* was taken at the same location 24 hours before the glider dive. The first profile is unaltered, subsequent vertical profiles are offset by 1 $^\circ$C or 0.1 g kg$^{-1}$ for clarity

## 2 Data collection and processing

### 2.1 Glider data

During January and February 2020, the RV *Meteor* surveyed the tropical North Atlantic, 200 km northeast of Barbados (Fig. 2), as part of the EUREC[4]A campaign (Stevens et al., 2021). During this cruise, the UEA glider group deployed three Seagliders. Seagliders are small buoyancy powered autonomous vehicles with wings that dive with a sawtoothed profile (Eriksen et al., 2001). They dive to a maximum of 1000 m with a vertical speed of around $0.1 \mathrm{~m~s}^{-1}$. In contrast to Argo floats, gliders are capable of manoeuvring to maintain position or to follow a survey pattern. Here, each glider dive limb is treated as a separate profile, so that a single descent-ascent cycle yields two profiles. The temporal overlap and dive depths of the three gliders are shown in Fig. 3. All glider dives were completed within the 10 km wide bowtie pattern shown in Fig. 2 except SG579, which was deployed 150 km south-west of the bowtie and arrived on yearday 36 (6 February). Gliders SG579, SG620 and SG637 completed 295, 131 and 155 dive cycles respectively, yielding a total of 1162 profiles for analysis, including 77 profiles from SG579 before it reached the bowtie. The gliders sampled every 5 seconds, for a vertical resolution of approximately 0.5 m. Owing to a shallow dive slope (where the glider has a small pitch angle relative to the horizontal), SG637 stalled during dives 22, 23, 41, 42, 43, and 47. Glider stalling reduces water flow through the unpumped CT sail, affecting temperature and conductivity measurements. We removed these dives from the dataset, as can be seen in the gaps in Fig. 3.

Glider profiles were processed using the UEA Seaglider Toolbox (Queste, 2014). This included tuning the hydrodynamic flight model following Frajka-Williams et al. (2011) and thermal lag corrections following Garau et al. (2011). After processing the glider profiles, temperature and salinity profiles were compared with CTD casts conducted by the *Meteor* (black vertical lines in 2.c). Quality controlled CTD profiles were provided by D. Baronowski (Baranowski, 2020). At the bowtie, SG637 dived to 750 m for 25 days, completing a dive approximately every 4 hours. SG579 only dived to 250 m and SG620 was recovered after 13 days. As SG637 recorded the most comprehensive set of observations, it is used for the majority of the subsequent analysis.

To apply the thermohaline staircases classifier to the dataset, we require data at regular depth intervals to ensure that features are consistently identified throughout the dataset. To achieve this, each profile was binned into 1 m depth bins using the median value of samples within the bin. When supplied with an even number of values, the function used (numpy.median) returns the average (mean) of the two middle values (NumPy Developers, 2022). Derived variables including the Turner angle, density ratio and vertical gradients in salinity and temperature were calculated at 1 m intervals using a 50 m vertical running mean to ensure smooth profiles, following the methods of VDB. A 50 m averaging interval is used as a smaller depth interval can be influenced by individual staircase steps (Shibley et al., 2017). Throughout this paper, temperature refers to conservative temperature, salinity refers to absolute salinity and density refers to potential density. All were calculated using the gsw toolbox implementation of TEOS-10 (Mcdougall et al., 2010; McDougall et al., 2011). Yeardays are used throughout the paper and start at 0 on the first of January 2020. Figure 2.c uses both yeardays and calendar dates for comparison.

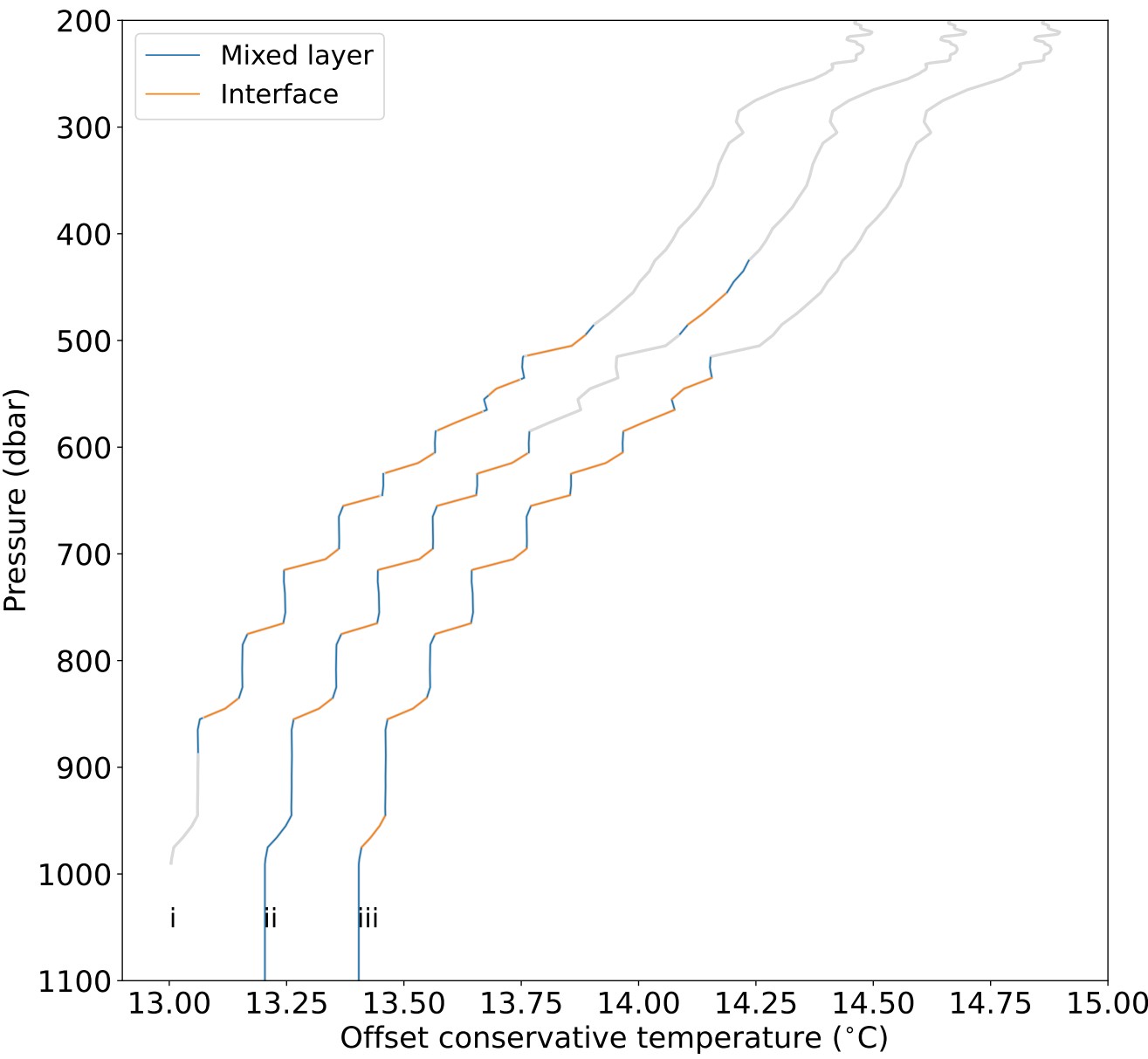

**Figure 5.** The same Argo profile shown in Fig. 1 from the Mediterranean classified three times. Three lines on this plot are (i) VDB's result, (ii) our classifier on default settings, and (iii) our classifier with a 10 % decrease in step height ratio and 4 % decrease in maximum permitted density gradient. The first profile is at the unaltered temperature, subsequent profiles are offset by 0.2 °C for clarity

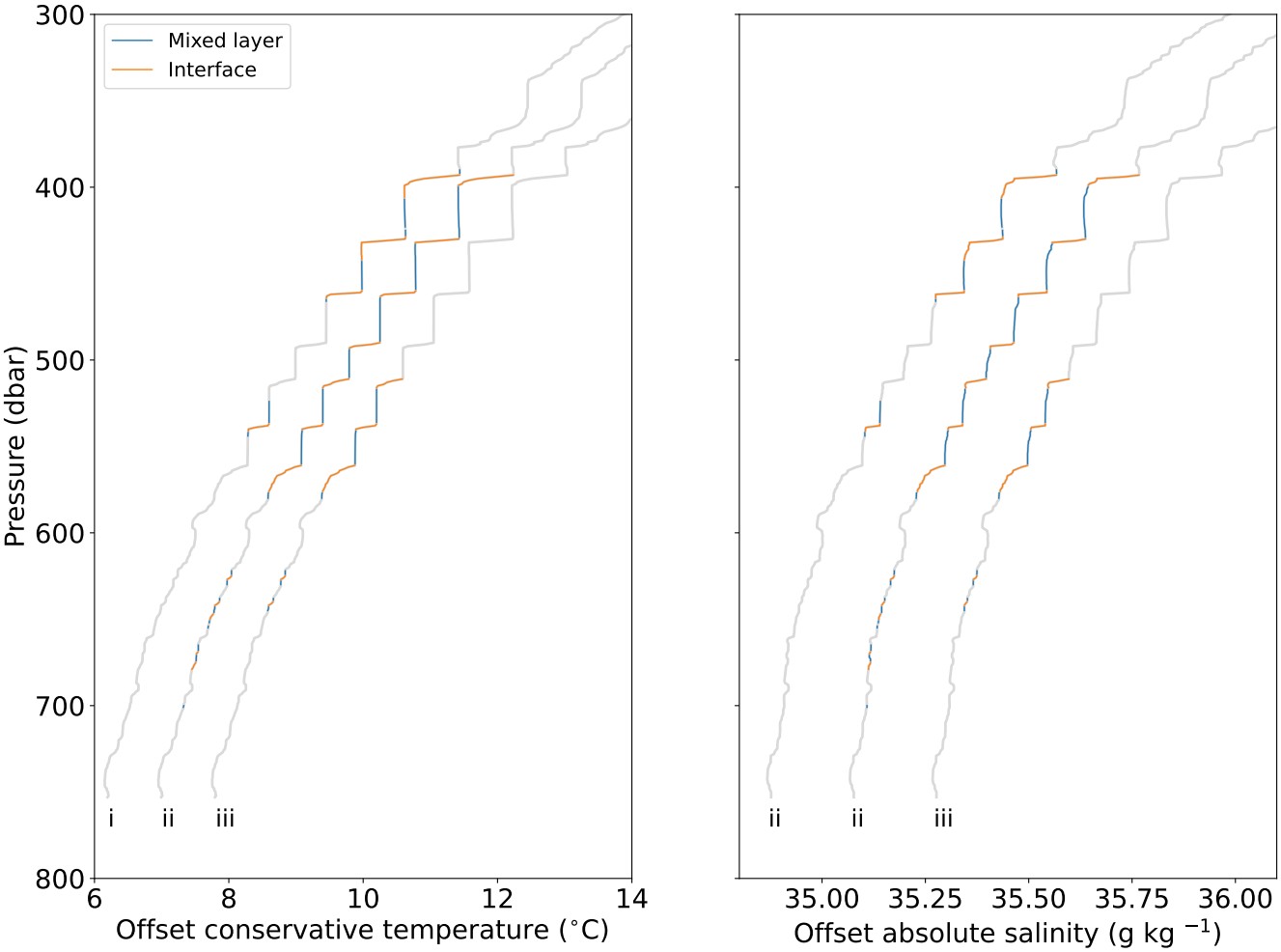

**Figure 6.** Classification of a glider profile from the Tropical North Atlantic demonstrating the differing output of VDB and our classifier. The three profiles are VDB (i), our classifier when operating on temperature only preliminary classification (ii) or temperature, salinity and density preliminary classification (iii). The profile is from dive 119 by SG637, yearday 41.6 (12 February). The first profile is at the unaltered temperature and salinity, the subsequent profiles are offset by 0.8 °C (a) and 0.2 g kg$^{-1}$ (b) for clarity

.

## 2.2 Thermohaline staircase classifier

To evaluate thermohaline staircases in glider profiles, we wrote a classifier following VDB. VDB's detection algorithm, identifies thermohaline staircases in a CTD profile using five steps:

1. Parts of the profile with vertical temperature, salinity and density gradients below a threshold value are classed as mixed layers.

2. Sections between mixed layers are preliminarily classed as interfaces

3. Interfaces are excluded if they exceed a maximum thickness, exceed the step height ratio, or exhibit less variability in salinity and temperature than neighbouring mixed layers. Step height ratio is the ratio of mixed layer height to the height of adjacent interfaces. Mixed layers that are smaller than step height ratio multiplied by adjacent interface height are discarded by the classifier. These processes ensure that mixed layers are separated by thin, steep interfaces.

4. Interfaces are classified into a double diffusive regime: salt-finger if the overlying mixed layer is warmer and saltier than the underlying mixed layer, diffusive-convective if the overlying mixed layer is cooler and fresher than the underlying mixed layer.

5. Staircases are identified as uninterrupted series of steps in the same double-diffusive regime. This step removes thermohaline intrusions. This step requires both salinity and temperature data, but uses only the average values of each mixed layer, so noisy salinity data can be used.

VDB's algorithm was designed to classify a global database of Argo profiles. It performs this duty well, but was specifically written to ingest Argo profiles at 1 dbar vertical resolution. We wished to make some modifications to the classifier including
120 testing its sensitivity to varying vertical bin size; this is not possible with the original algorithm. To identify staircases in our dataset, we built our own classifier (Rollo, 2021a) following the steps outlined in VDB, with the following adaptations:

- Able to process profiles at any regularly spaced step in pressure or depth.

- Optional user-defined maximum interface height. This sets the threshold for maximum distance between mixed layers that can be considered part of a continuous staircase.

- Optional plotting to show which layers are identified and discarded at each of the steps outlined by VDB in Sect. 3.

- Option to classify step shape using only vertical variation in temperature, rather than temperature, salinity and density. This is useful if salinity data are poor or recorded as a lower frequency than temperature, as is the case when using a fast-response thermistor.

- Each processing step as a separate, documented function.

- Software tests and validation against synthetic and field data.

- Software published in version-controlled repository (GitHub) to retain full development history. This includes a computational notebook of worked examples of the classifier.

These adaptations were driven by a need for greater flexibility in the classifier, and requirements imposed by the glider profiles. Substantial salinity spikes were present in the salinity profiles from all three gliders, because the time lag between the temperature and conductivity sensors has been incompletely removed for a region with such high vertical gradients (Fig. 4.b orange line). This is a common issue with the unpumped CT sensors used by these gliders. To remove these spikes, a 5 m running mean moving average filter was applied to the salinity profiles fed to the classifier (Fig. 4.b, green line). This removed the spikes, but smoothed the edges of the steps that the classifier identifies. To resolve this problem, we made the classifier able to identify the shape of steps using only temperature profiles. In this mode, salinity profiles are used to classify the staircase into the salt finger regime or the diffusive-convective regime. The classifier includes software tests to ensure that this use of temperature only for preliminary classification does not change the identification of staircases.

There were no Argo floats in the vicinity when our gliders were deployed, so we could not compare the output of our algorithm and VDB's algorithm on float profiles during the EUREC$^4$A campaign. Instead, we applied our classifier to the Argo profile used as a demonstration of VDB's algorithm by VDB Fig. A2. The results of this comparison are shown in Fig. 5 and discussed in Section 3.1. The geographic locations of data, and their sources, are detailed in Table 1.

An example of preliminary classification by temperature only is shown in Fig. 6. In this figure we contrast the outcome of our algorithm when the preliminary identification of steps uses temperature only (ii) or temperature, salinity and density (iii). In both cases, steps are subsequently classified into double diffusive regimes, and features such as thermohaline intrusions are rejected, using both temperature and salinity profiles. The benefit of classifying interfaces and mixed layers using solely temperature during the preliminary step, is that rounded step edges from the smoothing of salinity spikes, like those in the large steps around 400 - 500 m, do not prevent the identification of steps. To evaluate the effect of using only temperature profiles for the preliminary classification of steps, we processed the dataset described in this study using both temperature only and temperature-salinity-density preliminary classification and calculated summary statistics (Tables 2 and 3). These summary statistics show that the steps identified by both temperature only and temperature-salinity-density preliminary classification are very similar, both in their vertical extent and in the magnitude of change in temperature and salinity across the identified interfaces. The only substantial difference is the number of steps identified by each method. On average, profiles classified using solely temperature contained a median of 15 steps, over twice as many as when the same profiles were classified using temperature, salinity and density (7 steps). This is due to noisy salinity data incorrectly discarding steps during the preliminary classification.

The classifier is not without drawbacks. Notably, both temperature only and temperature-salinity-density classification methods fail to identify some smaller steps, such those around 620 - 640 m in Fig. 6, that have interfaces around 1 m thick, i.e. the resolution of the data fed into the classifier. They are visible to the eye, but fall below the identification threshold of the classifier. Preliminary identification of steps using solely temperature still requires salinity data that are sufficiently accurate to identify thermohaline intrusions and to classify staircases into the correct double diffusive regime.

| Figure | Data source | Location |
|---|---|---|
| 1, 5, | Argo float | Mediterranean |
| 3, 6, 7, 8, 10, 11, 9, 12, 13 | Glider data | Tropical North Atlantic |
| 2, 4 | Glider and ship CTD data | Tropical North Atlantic |

**Table 1.** Sources of the data displayed in figures.

| Statistic | Temperature only | Temperature-salinity-density |
|---|---|---|
| Total profiles | 1162 | 1162 |
| Profiles > 300 m | 682 | 682 |
| Profiles > 300 m with at least one step | 666 (97.5 %) | 652 (95.6 %) |
| Mean number of mixed layers per profile | 14.3 | 6.7 |
| Median number of mixed layers per profile | 15 | 7 |

**Table 2.** Summary statistics on all profiles. Total profiles is the number of profiles classified. Total profiles > 300 m is the number of profiles where the glider profiled deeper than 300 m. The mean and median number of mixed layers per profiles are the average number of mixed layers identified per profile. The two columns show these statistics when preliminary classification was conducted with only temperature profiles, and when preliminary classification was conducted with temperature, salinity and density profiles.

Our classifier identified 14205 thermohaline steps in the glider profiles. The glider dataset comprised 1162 vertical profiles recorded over 56 glider days. To make quantitative comparisons across the dataset, we use bulk statistics to analyse how the staircases changed over time. We define total thickness of mixed layers to be the total vertical extent of the profile identified as a part of a mixed layer in the thermohaline staircase, as defined in Fig. 1.a. This total mixed layer thickness does not include the interfaces. Mean mixed layer thickness is similarly calculated on a per-profile basis. Calculating these bulk values enables us to evaluate the temporal evolution of thermohaline staircases, and visualise datasets too large to plot directly.

## 3 Results and Discussion

### 3.1 Staircase Statistics

Using temperature only preliminary classification, thermohaline steps were identified in 97.7 % of profiles that extended below 300 m. In profiles with at least one mixed layer, the mean number of mixed layers per profile was 14.3. These statistics are summarised in Table 2. Coherent series of large steps were more limited, both temporally and spatially. Staircases with more than three mixed layers greater than 20 m were only observed at the bowtie, and only on two occasions: Yearday 23-25 (Fig. 3.b, 24-26 January) and yearday 40-44 (Fig. 3.d, 10-14 February). The changing thickness of steps is represented in Fig. 7 which shows the two events of strong staircasing as peaks in mean mixed layer thickness and total mixed layer thickness. The

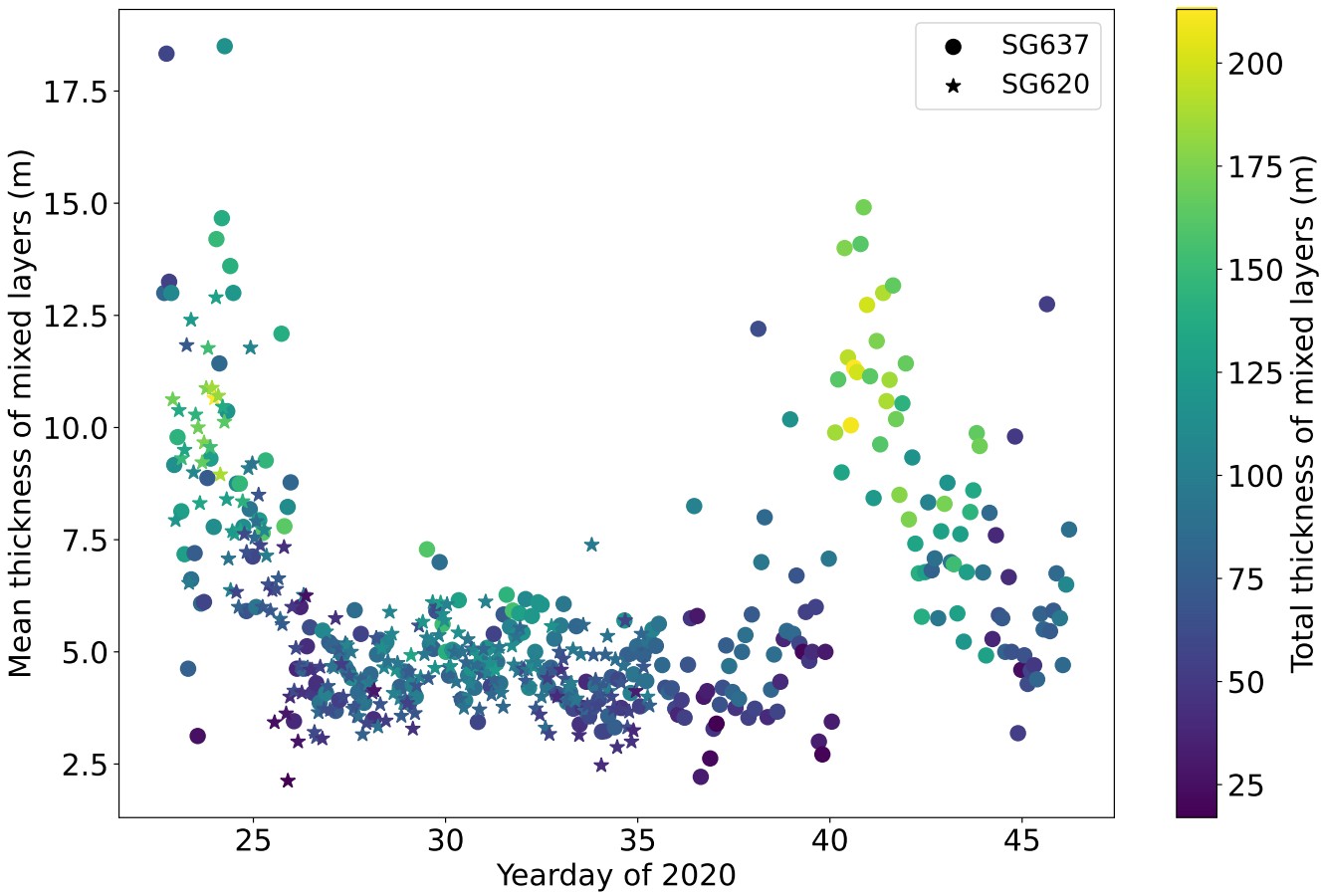

**Figure 7.** Mean thermohaline staircase mixed layer thickness observed by SG620 (stars) and SG637 (circles) coloured by the total thickness of the identified staircase mixed layers summed over the whole profile.. Glider data from the Tropical North Atlantic.

| Statistic | Temperature only | Temperature-salinity-density |
|---|---|---|
| Depth | 277-858 m | 300 - 766 |
| Temperature change across interface | 0.01-0.70 $^\circ$ C | 0.01-0.69 $^\circ$ C |
| Salinity change across interface | 0.001-0.12 g kg $^{-1}$ | 0.001-0.11 g kg $^{-1}$ |
| Interface thickness | 1-12 m | 1-14 m |
| Mixed layer thickness | 1-21 m | 1-19 m |

**Table 3.** Statistics on staircases. Ranges quoted are between the 2.5 and 97.5 percentiles, following VDB. Depth is the depth range over which staircases are identified. Temperature and salinity change across interfaces is the magnitude of change in these properties across the interfaces between adjacent mixed layers in a staircase.

small differences between values recorded by SG637 and SG620 can be attributed to the distance between the two gliders,
typically $\approx 5$ km (Fig. 2).

The incidence of thermohaline staircases varied spatially, with stronger staircases identified to the east, at the glider bowtie. SG579 observed few steps at its deployment location. As SG579 travelled eastward toward the bowtie larger staircases were encountered and the minimum observed depth of staircases decreased. A front was observed by SG579 as the glider passed over the continental slope on yearday 34 (4 February). Decreases of 0.5 $^{\circ}$C and 0.2 g kg$^{-1}$ in the uppermost 3 m coincided with a peak in chlorophyll a fluorescence and a broad along-slope current of 15 cm s$^{-1}$. Once past the front and over the abyssal plain, SG579 observed larger and more coherent staircases similar to SG620 and SG637 (Fig. 3.a). Total mixed layer thickness observed by SG537 increased from $58 \pm 25$ m to $90 \pm 26$ m after yearday 34 (4 February) as it approached the bowtie.

SG637 observed a sloping front later in the mission. First observed at 200 m on yearday 35 (5 February), the front reached 750 m, the maximum dive depth of the glider, on yearday 40 (Fig. 8 a&c, 10 February). The passage of this front was followed by the second period of large, coherent staircases at 400-600 m, with total mixed layer thickness increasing from $60 \pm 23$ m to $146 \pm 38$ m in the three days following yearday 40 (10 February).

Using the observations of all three gliders, we calculated bulk statistics following VDB (Table 3). The smallest features resolvable within our dataset are 1 m thick. 73 % of the steps identified were in the salt-finger regime 0.3 % were in the diffusive-convective regime, the reminder were not identified in either regime. Mixed layer thickness varied with $R_\rho$ (Fig. 9), the largest staircases were observed around $R_\rho = 1.7$, with a thick tail of steps at larger values of $R_\rho$ (Fig. 9).

Staircases in the salt finger regime require a density ratio between 1 and 2.5 (Fig. 10a). This constraint alone does not explain the distribution of thermohaline staircases in the glider profiles we collected. The absolute vertical gradients of temperature and salinity (Fig. 10.b & c) partially match the observed incidence of staircases in this dataset. The vertical gradients are calculated over 50 m to prevent individual thermohaline steps from affecting the results, in the same manner as the density ratio is calculated (Shibley et al., 2017). Using the critical values of density ratio, as well as upper limits of temperature and salinity gradient, we construct masks of where we expect thermohaline staircases to form. These masks are shown in Fig. 11, with a combined mask of the area that matches all criteria in Fig. 11 d. These masks suggest that staircases with steps larger than 1 m do not form where the absolute vertical gradients in temperature or salinity exceed 0.05 $^{\circ}$C m$^{-1}$ or 0.005 kg m$^{-1}$ respectively.

The results from our classifier are robust to changes in the key parameters, and relative differences within the dataset are preserved. However, parameter changes can affect classification of individual profiles. Figure 5 shows the result of three classifications of the same same Argo profile. The first profile is VDB's algorithm on default parameter settings, the second is our classifier using the same parameters, the third is our classifier with the maximum density gradient increased by 4 % and the step height ratio decreased by 10 %. In this example, three steps with less well defined edges are rejected by our classifier using default parameters (ii), as the relatively thicker mixed layers exceed the density gradient criterion. By making critical parameters more strict in profile iii, we achieve a closer, but not exactly matching, result to VDB's algorithm (Fig. 5).In this study we used the following key parameters: maximum mixed layer density gradient 0.001 kg m$^{-4}$ , maximum mixed layer density difference 0.01 kg m$^{-1}$, vertical averaging window for background profiles 100 m, maximum interface height 30 m. As

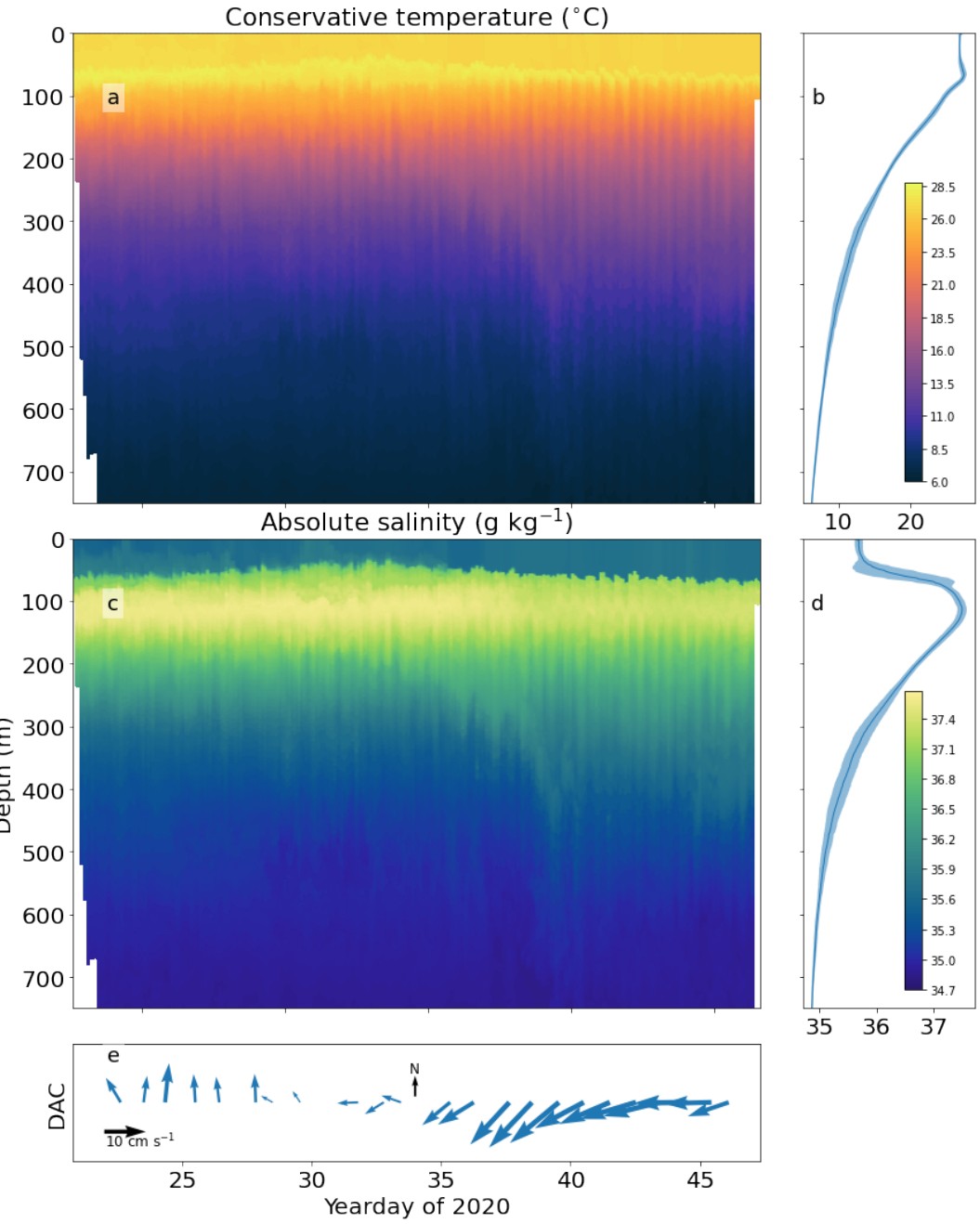

**Figure 8.** (a) Gridded mean conservative temperature from SG637. (b) Mean conservative temperature profile with uncertainty of 1 standard deviation shaded. (c & d) as (a & b) for absolute salinity. (a & c) gridding is 1 m vertically performed on a per-profile basis. (e) Daily mean of the dive average current (DAC). Glider data from the Tropical North Atlantic.

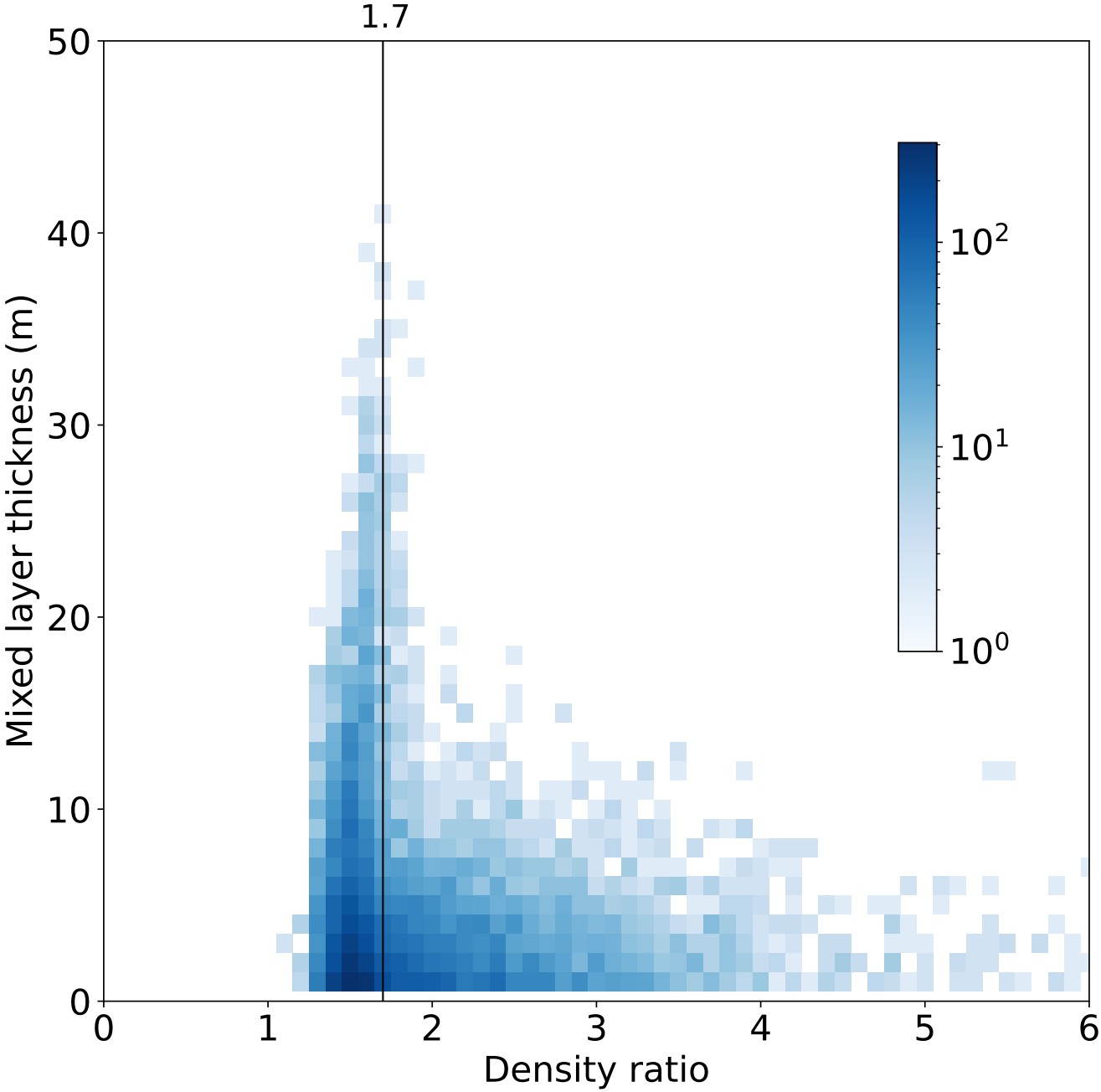

**Figure 9.** Observations of mixed layers classified from all three gliders between 100 and 800 m depth. Colourbar shows number of observations in each grid cell. Cell size of 0.1 in density ratio, 1 m in mixed layer thickness. Vertical black line marks the density ratio of 1.7. Glider data from the Tropical North Atlantic.

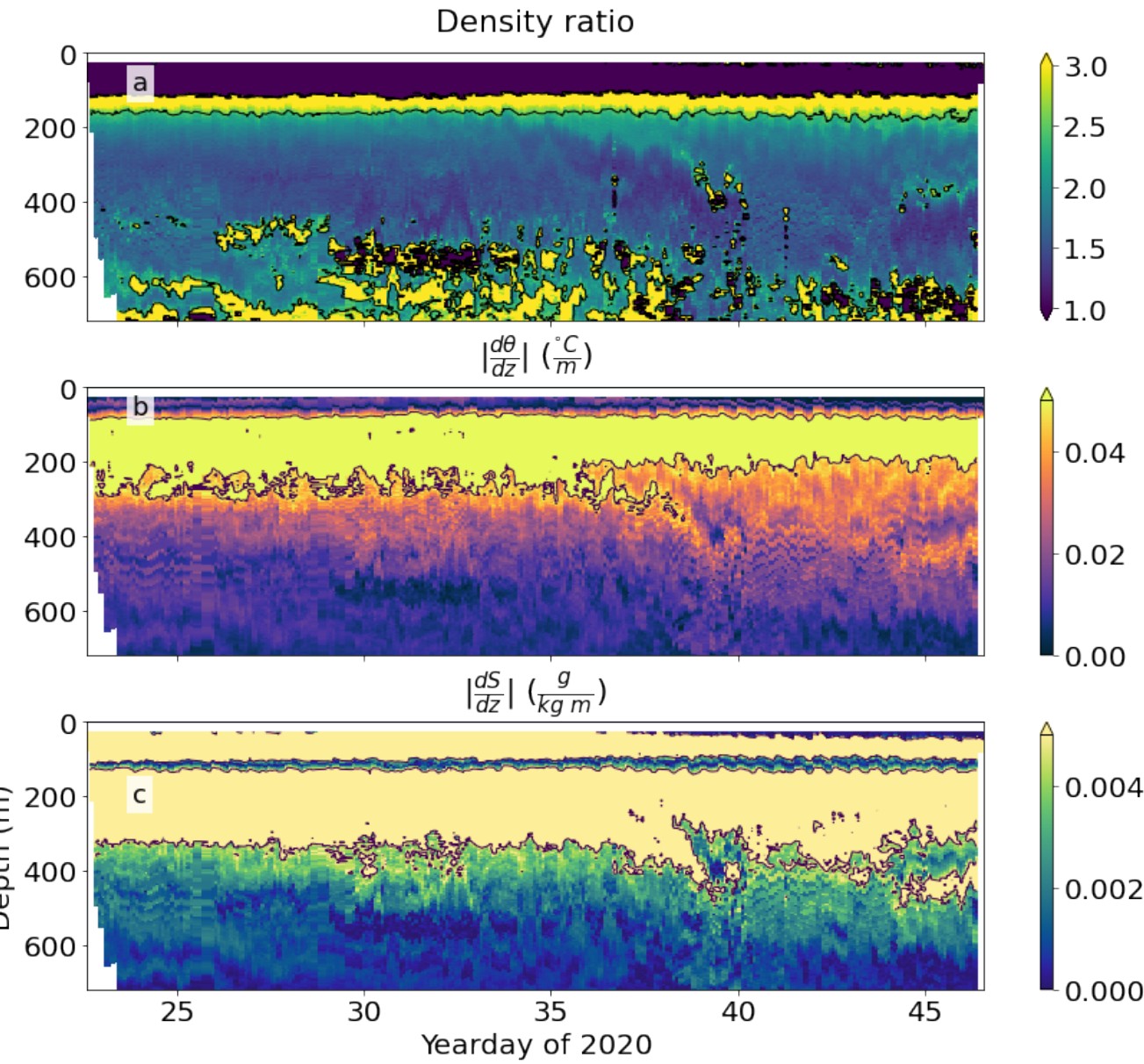

**Figure 10.** Key variables for the formation of thermohaline staircases, calculated from SG637 profiles. a) density ratio b) absolute vertical temperature gradient c) absolute vertical salinity gradient. Glider data from the Tropical North Atlantic.

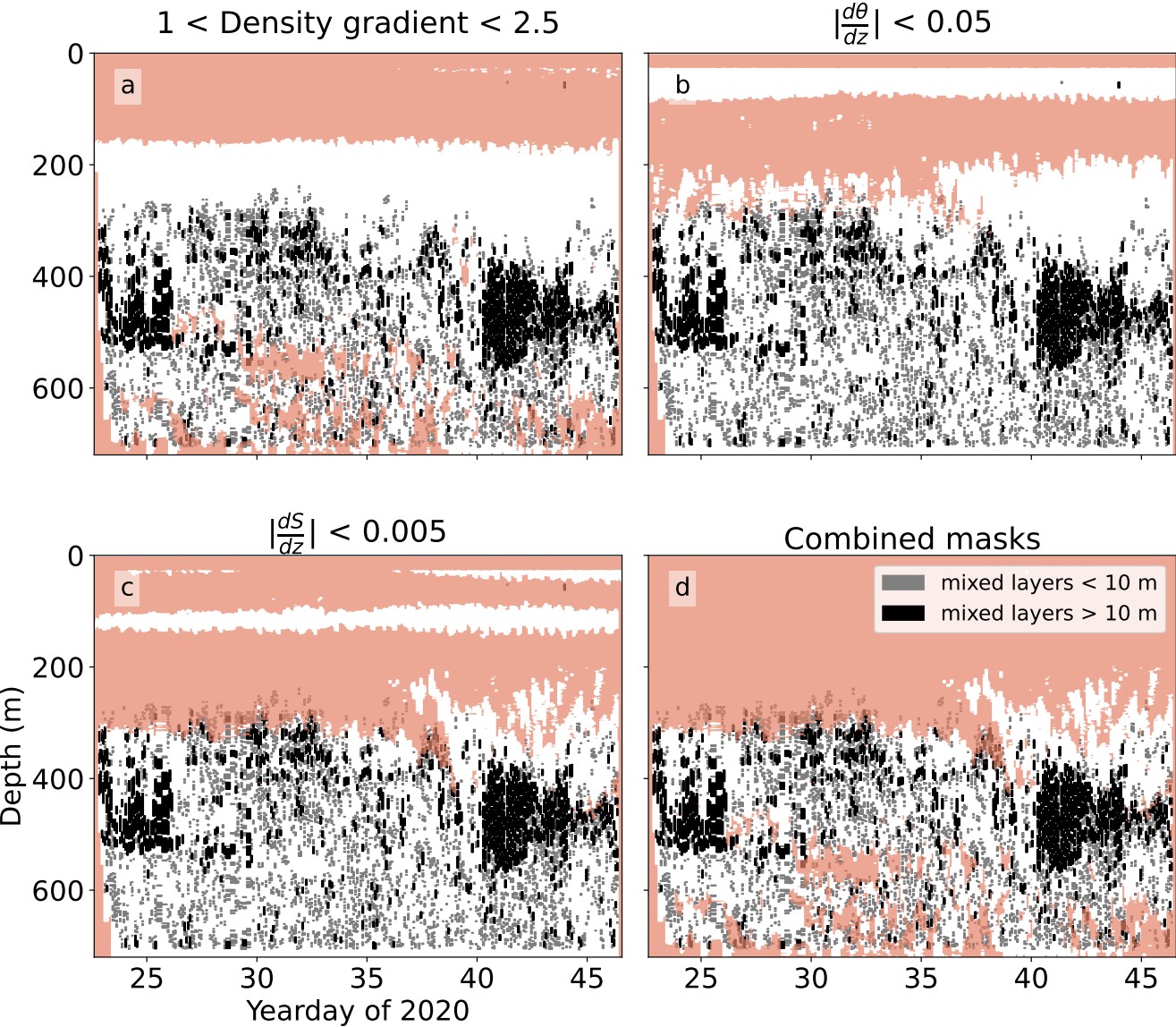

**Figure 11.** Masks shading the conditions in Fig. 10. Red areas mask where the conditions are not met. Grey blocks are the location of thermohaline mixed layers, black blocks are mixed layers thicker than 10 m. Glider data from the Tropical North Atlantic.

described in Section 2.2, we used solely temperature profiles during the preliminary classification step. The selection of these
215 key parameters is currently a manual process, guided by previous studies that have established threshold values (Bryden et al., 2014; Timmermans et al., 2008) and visual inspection of the resulting classification. Once chosen however, key parameters are constant for the whole dataset and the classification of staircases is not altered by the user post-classification.

## 3.2 Limitations of automated classifiers

To our knowledge, this is only the second study to use an automated classifier to identify thermohaline staircases, and the first to use one on glider profiles. Early studies with relatively few observations classified staircases manually (Schmitt et al., 1987; Marmorino et al., 1987; Ferron et al., 2021). Authors sometimes added further detail by classifying staircases as "clean " or "steppy" (Fer et al., 2010). Though more recent work preliminarily identified staircases by threshold gradients e.g. Durante et al. (2019, 2021), this was followed by inspection and reclassification by a human operator, again using descriptive labels such as "steppy", "rough" and "slope". Automated classifiers can remove the subjective element from this process, identifying staircases from their physical properties alone in a way that is repeatable and consistent within and between datasets.

The detection of staircases is dependent on the algorithm design and a small number of key parameters, such as maximum allowable density gradient, density variability and thickness of mixed layers and interfaces. VDB carried out sensitivity analysis on parameters including maximum mixed layer density gradients, temperature and salinity variability, and interface thickness. Their analysis showed that VDB's algorithm is most sensitive to changes in the maximum mixed layer density gradient and mostly insensitive to other parameters. Smaller maximum density gradients allow only the sharpest, most clearly defined steps. Larger gradients also identify the less well defined rough steps. VDB did not perform sensitivity analysis on vertical bin size due to the limited nature of Argo profiles which are supplied at 1 dbar resolution. Analysis on the effect of vertical bin size, as described in Sect. 3.5, suggests that this is an important factor in staircase identification. Using reanalysis data from the Mediterranean, Meccia et al. (2016) were unable to construct maps of staircases as the vertical scale of the reanalysis data was similar to the scale of the staircases. The scale analysis of thermohaline classifiers described in this section illustrates the importance of fine-scale sub-metre observations to classify thermohaline staircases.

Imperfect data also pose a problem for automated classifiers. Small deviations in temperature and salinity that a human classifier would ignore can cause steps to be rejected (e.g. Durante et al. (2019) Fig. 3). The nine filtering steps in VDB's algorithm and our classifier prevent most false positives, such as the step like structures formed by thermohaline intrusions. However, some visually step like structures are rejected if they fall outside the parameter space (Fig. 6). At the other extreme, setting key parameters to large values can allow overly lax step classification, where small interface layers between two mixed layers are not identified. This problem is most apparent at greater depths. In the Mediterranean, where staircases extend below 1500 m, individual interfaces have small magnitude changes in temperature and salinity. These small steps are harder to positively identify, particularly if there is instrument noise in the temperature and salinity profiles. Using VDB's default parameter values on a profile from the Mediterranean, our classifier combines two mixed layers around 1000 dbar (Fig. 5ii). Applying a 4 % decrease in maximum permitted density gradient and a 10 % decrease in step height ratio correctly identifies these as two separate steps (Fig. 5iii). VDB restricted classification to 100-900 dbar, so the performance of VDB's algorithm at this depth is unknown. Future classifiers could resolve the issue of identifying steps of smaller magnitude in deeper water by using varying parameters, perhaps scaling the threshold values for step identification by the background gradients in temperature, salinity and density.

### 3.3 Thermohaline staircases

The structures we observed during periods of strong staircasing are very similar to those encountered during the C-SALT campaign. Schmitt et al. (1987) recorded mixed layers of 5-30 m thickness separated by gradients with temperature and salinity jumps of 0.5-0.8 C and 0.1-0.2 psu in the area identified as strong staircasing. The glider bowtie is at the edge of this region (Fig. 2). This may explain why we did not observe coherent staircases throughout our observational period. Fer et al. (2010) also conducted their observations at the edge of the strong staircasing zone identified by C-SALT and observed intermittent staircases. The location of our observations at the edge of the staircase zone would also explain the slightly smaller step sizes and temperature and salinity changes observed compared with the averages reported by Schmitt et al. (1987). The rapid apparent breakdown of staircases over a day confirms the observations by Marmorino et al. (1987) of thermohaline layers disappearing within a few km. Due to the sampling pattern of the gliders, we cannot definitively determine whether this rapid transition from staircase to non-staircase is a temporal breakdown in the structure or the advection of the edge of the staircase zone past the glider bowtie.

Density ratio is generally assumed to be the main factor controlling thermohaline staircases formation and persistence. In agreement with previous studies in this region and elsewhere (Schmitt et al., 1987; Fer et al., 2010), we observed the largest steps around $R_\rho$ = 1.7 (Fig. 9). However, we also observed mixed layers at substantially larger $R_\rho$, though the maximum thickness of these layers was smaller, as predicted by Radko (2005). $R_\rho$ alone cannot account for the incidence of staircases, since we do not observe staircases shallower than 250 m, despite the favourable density ratio (Fig.s 10a and 11a). Similarly, Schmitt et al. (1987) did not observe any staircase structure at the $R_\rho$ minimum around 200 m depth, concluding that $R_\rho < 1.7$ is a necessary but not sufficient condition for the formation of staircases.

We propose that strong background vertical gradients in temperature and salinity prevent the formation of thermohaline staircases. As shown in Fig. 10, the water column at 200-250 m has a density ratio conducive to staircasing, but also has large gradients in temperature and salinity. Combining masks of all four key variables creates a better predictor of where staircases of large steps will form, particularly those with steps greater than 10 m in thickness (Fig. 11). This lack of steps in the upper pycnocline, also reported by Schmitt et al. (1987) & Fer et al. (2010), could also be due to observational limitations. In a manner analogous to internal waves, large steps form where background gradients in temperature and salinity are small (Fig. 5). This is most apparent in the Mediterranean, where the largest steps of several hundreds of metres are found deep in the water column, where background gradients are very small (Buffett et al., 2017; Durante et al., 2021). Where background gradients are strong, we would expect small thermohaline steps, possibly below the resolution of our 1 m binned profiles.

### 3.4 Wider implications

Predicting the presence of thermohaline staircases from background gradients can have practical benefits for grid-scale parametrisation. Thermohaline staircases in the tropical North Atlantic can increase diapycnal mixing rates by a factor of 5 (Schmitt et al., 2005). Previous studies have shown that thermohaline staircases occur mostly in well-defined regions (van der Boog et al., 2021a). Building on previous work, this study suggests that a set range of vertical temperature and salinity profiles are

conducive to the formation of thermohaline staircases. Where these characteristic profiles are present in models, differing vertical diffusivities of heat and salt can be applied to parameterise the enhanced vertical diffusivities caused by double-diffusive processes. Most ocean models lack sufficiently fine vertical resolution to directly represent thermohaline staircases comprised of mixed layers $\approx 10$ m and gradient layers $\approx 1$ m in the open ocean environment where thermohaline staircases are most often found. The enhanced diapycnal mixing driven by staircases could be parametrised where key conditions are met, removing the need for direct representation of staircases.

The acoustic impedance contrast across a thermohaline step is an ideal reflectors for seismic oceanography. Previous studies (Fer et al., 2010; Buffett et al., 2017) have successfully exploited this characteristic to map out thermohaline staircases over large spatial scales. Often, these 3D maps of staircases are constructed from seismic data and XBTs. Gliders can ably complement the analysis of seismic data. Gliders are particularly valuable for their ability to maintain station or sample a specific region distant from the ship conducting the seismic survey, unlike free floating Argo floats or XBTs which are single use and must be deployed at the required site by ship or aircraft. Seagliders are a useful tool for observing thermohaline staircases in the tropical North Atlantic as the staircases are limited to the uppermost 800 m, well within Seagliders' 1000 m maximum depth rating. As opposed to the Tyrrhenian Sea where staircases extend deeper than 1500 m (Buffett et al., 2017). This study has proven the usefulness of gliders in this region. Future seismic oceanography studies in the tropical North Atlantic could benefit from the inclusion of gliders in a similar manner to Buffett et al. (2017).

Despite this is a study using with less than two months of observations, we have proved that gliders can be successfully used to identify thermohaline staircases with an automated classifier. More work should be done to test the hypothesis of strong vertical gradients in temperature and salinity limiting the incidence of large thermohaline staircases. It would be beneficial to test if observations in other staircasing regions such as the Arctic and Mediterranean exhibit similar behaviour. In particular, staircases in Arctic regions dominated by diffusive-convective processes typically have smaller steps than those in the salt finger regime (VDB). Shibley et al. (2017) used Ice-Tethered Profiler data at $\approx 25$ cm vertical resolution to identify staircases in this region. A high-resolution glider dataset from the Arctic could add to this body of work, revealing more small scale staircases than Argo floats are capable of resolving. This work could extend under the ice if acoustic navigation of gliders is adopted.

### 3.5 Scale sensitivity of the classifier

One of the key variables not tested by VDB is the vertical spacing of observations within profiles. We took advantage of a high-resolution temperature sensor to test the classifier's sensitivity to varying vertical spacing. In addition to a CT sail, SG620 was equipped with a Rockland Scientific microstructure system, including an FP07 fast-response thermistor. The fast-response thermistor sampled at 512 Hz with a response time of ~ 10 ms, measuring temperature at a spatial resolution of ~ 1 mm. We binned the temperature profiles at 0.2, 1 and 5 m to test the effect of vertical bin size on the classifier. Each binned value is the median of all samples within the bin. Even at 0.2 m bin size, each bin average is calculated from over 100 measurements. The classifier initially identified steps based solely on temperature structure. Salinity values taken from the nearest CT sail measurement were used for subsequent classification into the correct double diffusive regime. An example profile at the three

binning intervals is shown in Fig. 12. The gaps between adjacent mixed layers and interfaces are clear in the 5 m binned profile, as a point cannot be part of both the mixed layer and an interface (Fig. 12b).

When classifying profiles, the three bin sizes give similar results. The differences are most obvious at the upper edge of the staircase zone, where individual steps are smaller (Fig. 12c). The 5 m binned profile completely erases the ~ 2 m thick steps. The 1 m binned profile also misses the steps, as the ~ 50 cm thick interfaces contain at most 1 point. Our classifier requires two points in the interface to classify it. This sets a minimum layer thickness equal to twice the bin size, a limitation shared with VDB's algorithm. A separate issue is apparent with finely binned profiles. Using a smaller bin size, the 40 m thick step around

420 m is excluded (Fig. 12b). This is due to the increased variability, introduced by the extra points in the profile, exceeding the maximum allowable temperature variation and causing the step to be discarded. Reconciling these behaviours will be a challenge for future iterations of the classifier. Perhaps a more sophisticated moving average filter than the simple boxcar used here could smooth out small perturbations in temperature while retaining the sharp shape of the staircases.

Considering the total results from all of SG620's dives (Fig. 13) it is clear that finer binning (blue dots) identifies more

layers at all depths and of all thicknesses. This is particularly apparent in the upper parts of the pycnocline around 100-200 m. In addition to missing shallow layers, the 5 m binned profiles detect no steps below 500 m, where the temperature changes of the interfaces are smaller. The distribution of density ratio (Fig. 9) is robust to changes in bin size, with a strong peak of thick mixed layers centred around $R_\rho = 1.7$. However, the 5 m binned profiles detected no steps at $R_\rho > 2.5$. This may explain why early studies such as Schmitt et al. (1987); Marmorino et al. (1987), with coarser vertical sampling resolution, did not

detect any staircases at large values of $R_\rho$, or in the upper reaches of the pycnocline. This also imposes a limit on the structure detectable from Argo data, which has a minimum detection threshold of 2 dbar (VDB). On bandwidth limited platforms like Argo floats and seal tags, this issue could be circumvented with on-board processing. Using recorded profiles of temperature and salinity at their maximum resolution, thermohaline staircases could be identified using an algorithm like VDB's algorithm and the location of steps transmitted over a satellite link. This would avoid the need to transmit the high resolution thermohaline

profiles while identifying steps at a finer resolution than 2 dbar.

Automated classifiers for thermohaline staircases are in their infancy. We hope that the example created by VDB and built on by this work can be further improved, and that other classifiers may be created to resolve issues raised in this section. We have made our classifier available on GitHub, with full development history, software tests, sample data and a demonstration notebook (Rollo, 2021a). Future work on thermohaline classifiers, and many other code intensive oceanographic projects, could

benefit from the iterative, collaborative approach of open source software development enabled by platforms like this.

## 4   Summary

Building on the work of VDB we developed a classifier to detect thermohaline staircases. This improved classifier can operate on datasets at any regular vertical spacing. The classifier can also be used on datasets with suboptimal salinity data, such as profiles contaminated by salinity spikes, or salinity measured at lower spatial resolution e.g. when using a fast-response

thermistor. We used this classifier to successfully identify two periods of strong thermohaline staircasing in the tropical North

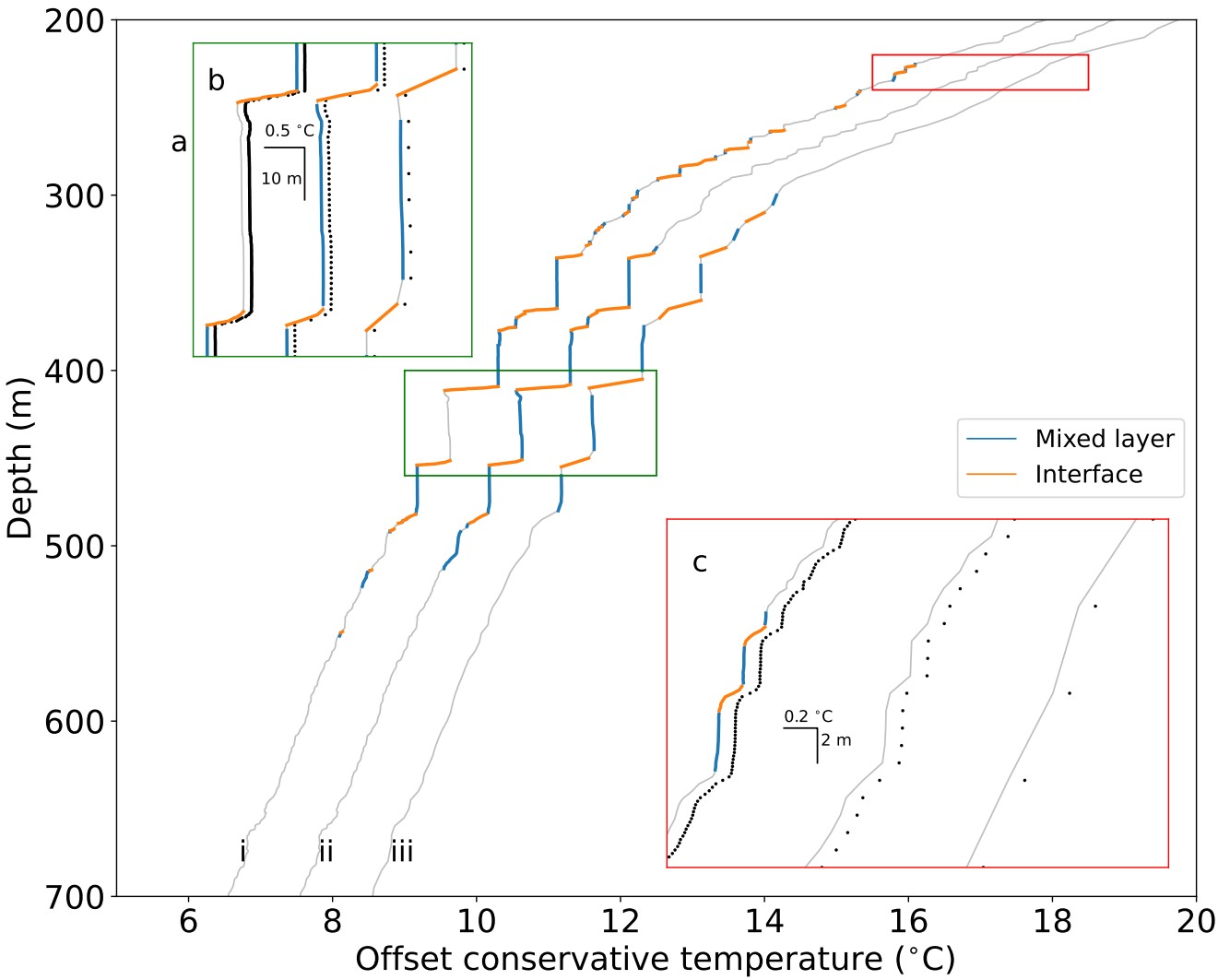

**Figure 12.** SG620 dive 20 fast-response thermistor temperature. Three profiles are the same data binned at 0.2 m, 1 m and 5 m respectively, then classified. b & c are detailed subsamples of the profiles, offset black dots mark the centres of each depth bin. Glider data from the Tropical North Atlantic.

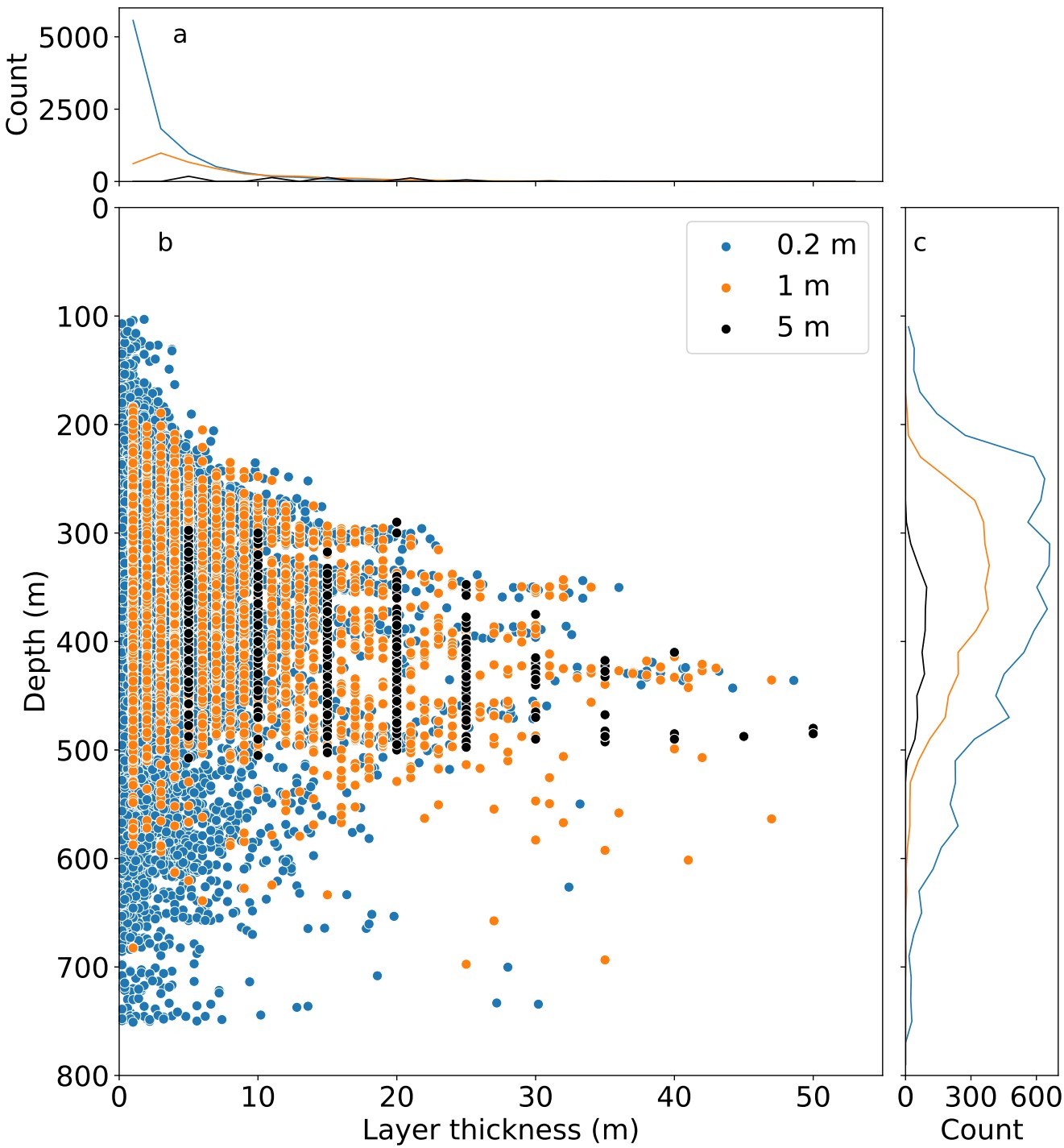

**Figure 13.** All mixed layers identified from SG620 fast-response thermistor data when binned at 0.2 m, 1 m and 5 m. Lines in a) and c) are binned total number of points in b) with bin widths of 2 m and 20 m respectively. Glider data from the Tropical North Atlantic.

Atlantic. We identified thermohaline staircases in 97.7 % of profiles that extended below 300 m. Our classifier is flexible, extensible and designed to be simple to use.

Using fast-response thermistor data, combined with salinity observations, we were able to classify staircases at the sub metre scale. This is a promising demonstration of automated classification at smaller scales, which can be useful in areas such as the Arctic where small scale thermohaline staircases are known to form. We also used these data to test the sensitivity of the classifier to varying vertical bin sizes, finding that datasets with large bin sizes substantially underestimate the incidence of thermohaline staircases.

Based on our observations of thermohaline staircases, we hypothesise that strong vertical gradients in temperature and salinity inhibit the formation of staircases in locations where the density ratio is favourable to their formation. Implementation of this result could improve model sub-grid parametrisation of diapycnal mixing in regions such as the tropical North Atlantic and Mediterranean, where thermohaline staircases are widespread and long-lived.

*Code and data availability.* Thermohaline classifier code is archived at Rollo (2021a). Glider data are held at the British Oceanographic Data Centre (Rollo, 2021b).

*Author contributions.* RAH, KJH and CR contributed to the design of the experiment and data collection. CR created the algorithm and produced the figures. CR prepared the manuscript with contributions from KJH and RAH.

*Competing interests.* The authors declare that they have no conflict of interest.

*Acknowledgements.* CR was supported by the Natural Environment Research Council and the Engineering and Physical Sciences Research Council, via the NEXUSS Centre of Doctoral Training in the Smart and Autonomous Observation of the Environment Grant NE/N012070/1. This project has received funding from the European Research Council (ERC) under the European Union's Horizon 2020 research and innovation programme (COMPASS, Grant agreement No. 741120). We thank the crew and scientists of the RV *Meteor* cruise M161 for assistance in the deployment and recovery of gliders, in particular Dariusz Baranowski for collection and processing of ship CTD data. Thanks to Gareth Lee and Marcos Cobas-Garcia for the preparation of the gliders, and the UEA glider group for piloting. Argo data were collected and made freely available by the International Argo Program and the national programs that contribute to it (Argo, 2021). (https://argo.ucsd.edu, https://www.ocean-ops.org). The Argo Program is part of the Global Ocean Observing System. We used the Python packages numpy (Harris et al., 2020) and pandas (McKinney, 2010) to create the classifier. We used matplotlib (Hunter, 2007), cartopy (Elson et al., 2020) and Jupyter (Kluyver et al., 2016) in the creation of figures.

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
