# Peer review of "Glider Observations of Thermohaline Staircases in the Tropical North Atlantic Using an Automated Classifier"

_Geoscientific Instrumentation, Methods and Data Systems, 2021_

## Referee Comment (RC2)

[referee-annotated manuscript omitted]

---

## Author Response (AR1)

**Response to reviewers**

We thank Carine van der Boog and one anonymous reviewer for their very helpful recommended improvements to this manuscript. We will incorporate all the reviewers recommendations in our revised manuscript.

**Response to comments**

The main point raised by both reviewers is our use of suboptimal salinity data for the classification of thermohaline staircases. In the revised version of the manuscript, we will better explain how our algorithm uses salinity and temperature profiles by adding additional sentences in Section 2.2 . As Reviewer 2 noted, our classifier uses temperature profiles to identify potential thermohaline staircases. Temperature and salinity profiles are used to distinguish between thermohaline staircases and other structures such as thermal intrusions. As part of the revision, we will include in Section 2.2 a quantitative analysis on the effect of using both temperature and salinity profiles for the initial step of identifying thermohaline staircases, as performed by Carine van der Boog et al. (2021), vs using only temperature profiles for this step.

Following Reviewer 1's detailed comments, we have made the following changes:

- More accurate consideration of the contribution of thermohaline staircases to the meridional overturning circulation, referencing recent literature (see later list of papers to include)

- More careful use of terms steps, interfaces, layers and staircases throughout the paper

- Remove duplicate discussion of Turner Angle, as this is adequately described by the density ratio

- Explanation of terms such as maximum mixed layer height and step height ratio

- More explanation of glider specific terminology such as glider slope, as well as a more thorough description of the glider microstructure system and differences between ascending and descending profiles

- Describe the binning methodology in more detail, particularly the result when a given bin contains only two samples.

- Expand discussion of the identification of steps in the salt-finger and diffusive-convective regime

- Discussion of previous work with ice tethered profilers in the Arctic, and how gliders could complement these observations.

- A clearer explanation of the selection and use of critical parameters for the detection of thermohaline staircases, including tables, and how these affect out results.

- Clearly stating which observations and conclusions are based on Mediterranean and Atlantic data.

- More quantitative comparison between the classifications produced by our algorithm and VDB operating on the same profiles.
- Checking the order of figures

Following Reviewer 2's recommendation, we make the following changes:

- Clarify our use of Mediterranean vs North Atlantic data in text and figures.
- Add more geographic detail to our analysis, particularly in the Section 2.1
- Explicitly state our use of units
- Add more discussion on the effect that varying vertical bin size has on the classification of thermohaline staircases in Section 4.2.
- Using calendar dates as well as yeardays to help orient readers in Section 2.1
- Increase the connections between text and figures in Section 3
- Discuss more of the previous studies within this domain that has been undertaken in the Mediterranean

**Annotated manuscript**

In their annotated manuscript Reviewer 2 echoed many of the points raised by Reviewer 1, and also made a number of useful recommendations to improve the manuscript. These recommendations included suggestions to standardise language, quote more specific numbers where available, highlight more salient areas of figures and make better use of colour. We have worked these helpful recommendations into the revised manuscript.

The reviewers brought several recent papers to our attention that we had not discussed in the original submission. We have included discussion of the following papers in our revised manuscript:

**van der Boog et al. 2021** Double-diffusive mixing makes a small contribution to the global ocean circulation https://doi.org/10.1038/s43247-021-00113-x

**Shibley et al. 2017** Spatial variability of the Arctic Ocean's double-diffusive staircase https://doi.org/10.1002/2016JC012419

**Durante et al., 2021** Mixing in the Tyrrhenian Interior Due to Thermohaline Staircases

https://doi.org/10.3389/fmars.2021.672437

**Ferron et al. 2021** Contribution of Thermohaline Staircases to Deep Water Mass Modifications in the Western Mediterranean Sea From Microstructure Observations https://doi.org/10.3389/fmars.2021.664509

**Meccia et al. 2016** Decadal variability of the Turner Angle in the Mediterranean Sea and its implications for double diffusion https://doi.org/10.1016/j.dsr.2016.04.001

**Community review**

This manuscript has not received any community reviews from non-nominated reviewers. However, Frederic Merceur of IFREMER contacted us via email to request a more complete acknowledgment of Argo data used in this study. We have added his suggested citation to the acknowledgements section of the revised manuscript.

**Itemised response to reviewers**

In this section, we reproduce the specific and line-by-line comments made by reviewers, and detail the changes we have made in response to these. Original review comments are included in blue, our responses are in black.

**Reviewer 1**

**Major comment:**

One of the suggested changes is that the algorithm can now use suboptimal salinity data. As thermohaline staircases have a staircase structure in both the temperature and salinity profiles, I wonder how this adjustment affected the performance of the algorithm. (how do you distinguish thermohaline staircases from thermohaline intrusions, etc.) Because the impact of this change is only discussed qualitatively, it remains unclear whether this simplification can be applied. So therefore, a quantitative discussion on this topic is necessary to justify this adjustment.

The section describing the adaptation of the algorithm was poorly explained in the original submission. When classifying by temperature, rather than temperature, salinity and density, only the initial step of identifying mixed layers and interfaces is changed to use solely temperature profiles. Salinity profiles are still used in identifying thermohaline intrusions. We have rewritten this section, and added a figure showing an example classification using our algorithm both in temperature only and temperature-salinity-density mode. We have also performed some quantitative analysis comparing the results of classifying our datasets in these two modes, this includes a two new tables of comparative statistics.

**Line-by-line comments:**

line 33-36: These two sentences suggest here that double-diffusive mixing affects the meridional overturning circulation. However, this effect seemed to be negligible small when considering the contribution of the mixing by thermohaline staircases (https://doi.org/10.1038/s43247-021-00113-x). Therefore, this suggestion needs to be either weakened or given more context.

Citation added to https://doi.org/10.1038/s43247-021-00113-x qualifying this statement and referencing the negligible contribution of thermohaline staircase associated mixing to the global ocean mechanical budget.

line 40: What do you mean with 'steps'? Is that the mixed layer thickness? Or do you refer to the height of the temperature and salinity steps between the mixed layers?

"Steps" has been replaced with "individual mixed layers" to improve clarity of this sentence

line 50: 'Firstly, the Turner angle must fall in the regime favourable to double diffusive processes of diffusion-convection ($-90 \leq Tu \leq -45$) or salt finger ($45 \leq Tu \leq 90$). Secondly, the density ratio must be

within a critical range.' The Turner angle and the density ratio are both indicators of the stratification, and can be transformed into each other with the equation: Rρ = −tan(Tu+45â¦). So, this means that the Turner angle and density ratio are in essence two different variables to describe the same thing. Therefore, it is not clear to me why the Turner angle and density ratio are described as two different criteria here.

We have removed the plots and discussion of Turner angle to focus solely on density ratio in the results and discussion sections. We have kept Turner angle in Fig 1. to illustrate how it relates to the density ratio.

line 72: Does the accuracy of the observations differ between ascending and descending profiles, due to the positioning of the sensors and the turbulent wake induced by the glider? And if so, how does that affect your results?

The turbulent wake of the glider is similar during ascending and descending profiles, as the vertical speed and dive angle are kept constant by the pilots(Eriksen et al., 2001). Considering the scale of staircases classified by this study, we would not expect the small differences in flight characteristics during descending and ascending profiles, like those shown by  Frajka-Williams et al. 2011, to affect out results

line 77: what is a shallow dive slope?

A shallow dive slope describes a situation the glider has a small pitch angle relative to the horizontal and is prone to stalling. This has been added to the text.

line 88: you mentioned that the glides have a vertical resolution of approximately 0.5m (line 77), and mention here that you binned the data into 1m depth bins using the median value of the samples within the bin. Does this mean there are usually only 1 or 2 points in each bin? If so, what do you define as the median of 2 values? Do you take the upper or lower value, or is this random?

Excellent point. When supplied with an even number of values, numpy.median returns the average (mean) of the two middle values (https://numpy.org/doc/stable/reference/generated/numpy.median.html).  This is the case when supplied with two values. Explanation and reference added to text

line 101: interfaces is more widely used than gradient layers. You can consider using interfaces instead.

Gradient layers replaced with interfaces throughout manuscript text and figures

line 108: 'diffusive-convective regime' should be 'double-diffusive regime'.

Corrected in the text

line 110: Which hard-coded aspects do you refer to here? I would assume that, because you binned the data into 1m depth bins (line 88), that you should be able to do a direct comparison with the original algorithm.

We have directly run the original algorithm against our glider data. We present the comparison of the algorithms in the new Figure 5. This sentence has been reworded "We wished to make some modifications to the classifier including testing its sensitivity to varying vertical bin size; this is not possible with the original algorithm."

line 115: Can you add an explanation why you added a 'maximum mixed layer height'?

This should have been "maximum interface height". This value sets the threshold for maximum vertical distance between mixed layers that can be considered part of one continuous staircase. Depending on the scale of the staircases being studied, this can be varied to correctly identify separate staircases in one temperature-salinity profile. We have added an explanation to the text

line 120-122: It might be helpful to add your documentation to the supplementary information (or appendix) for reference.

We have added notebook of examples usage and docstrings to the repo of the classifier, linked in the manuscript

line 135: does the total thickness of the mixed layers also include the interfaces. If so, isn't it then more a measure for the total height of the staircase?

The total thickness of mixed layers does not include interfaces. I have updated the schematic in Figure 1.a to better represent this and added a clarification on line 135.

line 140: Where do you show this? You could consider adding a table with these numbers (also numbers from 159-164) to clearly summarize your findings.

Tables added with summary statistics.

line 161: '73% of the steps identified were in the salt-finger regime'. Does this mean that the other 27% are in the diffusive-convective regime? Or are they not identified as either regime?

0.3 % were in the diffusive-convective regime, the reminder were not identified in either regime. This has been added to the text.

line 163: No steps were observed at R<1. Doesn't this directly follow from the requirement that the Turner angle of the steps should be in double-diffusive regimes?

Yes, sentence removed.

line 169: Because the haline contraction coefficient and thermal expansion coefficient vary over depth, the upper limits of the temperature and salinity gradients vary over depth in terms of their density contribution. Why did you use the temperature and salinity directly instead of their density components?

During initial analysis, the direct temperature and salinity gradients better matched the features we observed than their density components, so we opted to show plots of the gradients.

line 182-186: I do not completely understand why the latter explanation is more likely. Can you elaborate on that?

Removed assertion of the latter explanation being more likely.

line 188: You miss a reference here.

Citations added to Schmitt et al. 1987 and Fer et al. 2010.

line 195: Looking at 200-250m in Fig. 9, it seems that the masks of the Turner angle (Fig. 9a), the density ratio (Fig. 9b) and the salinity gradient (Fig. 9d) are all valid, while the temperature gradients (Fig. 9c) changes in time. Can you explain why you use all 4 criteria, instead of just the temperature gradient, as that appears to be the governing one.

We have removed the masks of Turner Angle, as these duplicate the information shown in the masks of density ratio. This enables us to show the mask of salinity gradient in addition to the temperature gradient. These masks contribute to different areas of the combined mask, which better match the region where we do not observe large scale staircases.

lines 200-203: It is not entirely clear to me on which data the results are based in different sections. For example, the glider data is obtained in the North Atlantic, while they are here compared to thermohaline staircases in the Mediterranean Sea. This is confusing, because Fig. 1 and Fig. 5 contain observations from the Mediterranean Sea.

To clarify the sources of data used in the figures, we have added a table as suggested by reviewer 2. This table describes the location and data source(s) of each figure. We have also added a sentence to each figure specifying the data source(s).

line 224: In the Arctic, the raw data of the Ice-Tethered Profilers can be used to analyze the thermohaline staircases. The vertical resolution of this data is much higher than the Argo floats. I think it is worth mentioning that there has been studies that analyzed the Arctic staircases (for example: https://doi.org/10.1002/2016JC012419), before discussing how gliders can be used in that region as well.

Discussion of Shibley 2017 added in this paragrapaph

line 226: 'varying vertical spacing'. Is this varying within a profile?

Yes. "varying vertical spacing" replaced with "vertical spacing of observations within profiles" for clarity.

line 274: What is a 'step height ratio'?

Step height ratio is the ratio of mixed layer height to interfaces height. Mixed layers that are smaller than step height ratio multiplied by adjacent interface height are discarded by the classifier. This ensures that classified mixed layers are separated by relatively thin interfaces. Explanation added to text in Thermohaline staircase classifier section.

line 279: 'We used the parameter set demonstrated in profile iii for this study'. What are exact numbers / settings that you used? And does that mean that you only used temperature profiles to detect the staircases throughout this study? How did that affect your results?

We have added the key parameters used in this study and confirmed that we used temperature profiles only in the initial classification. Discussion of the effects of this choice has been added to Section 2.2.

line 301: If your algorithm works on any regular vertical spacing, could you then apply it as well to the dataset of VDB? If so, you can make a comparison between the two algorithms and more quantitatively (compared to Figure 5) discuss how and where the detection of staircases differ. Such an analysis would also clarify whether using only temperature profiles give significant different results.

We have added direct comparison of the two algorithms on a profile from our glider dataset, as well as statistical comparison between the results of classification with our algorithm in temperature only and temperature-salinity-density profiles fro initial classification. This is described in more detail in our response to the major comment above.

line 305: can you elaborate more on how you can use your study to improve model sub-grid parameterizations?

Paragraph expanded with more sentences describing the potential for using background gradients in models to predict probable thermohaline staircases at sub-grid scales an adjust diapycnal mixing rates accordingly

**Other comments:**

the figures do not appear in chronological order.

This has been corrected.

**Reviewer 2**

**Specific comments**

If I have understood correctly you are not using salinity data in the classifier, except than for a double check and parameters calculation. If I am right, you shouldn't use the claim about the applicability of your classifier on data with weak salinity reliability, this would be true if you include salinity using some kind of enhancement on the data. Despiking is a necessary data processing step always, so if you still have so very bad salinity data and you do not trust them, you should just not use them and base your algorithm only on temperature (if you can). Actually, in a process like double diffusion, that is double indeed, salinity should be taken into account in a classifier algorithm, maybe with some warnings. I agree with Rev1 on the necessity of more wide clarification on this part.

We have rewritten this paragraph to explain how the classifier uses temperature to perform the initial classification, then uses both temperature and salinity to classify the double diffusive regime of the staircase. We have added a figure comparing the two methods, two tables of comparative statistics, and a clarification that the classifier requires sufficiently accurate salinity data to classify the staircases, even when the initial step uses only temperature data.

You are using North Atlantic data, but you also show Mediterranean data and cite Arctic data. This is a bit confusing, because it seems that you want to compare different kind of staircases, which is a good point (also because one of the strongest point of your classifier is its flexibility on different dataset), but if so the comparison need to be more structured in the text (maybe with specific paragraph or table?).

We have added a table denoting the data source and location of the data shown in each of the figures. Additionally, every figure now has a sentence denoting the data source(s)

The manuscript has different writing styles and this affects its fluency. In particular, introduction and some part of the result paragraphs are very broken up. It should be very much better if you balance and harmonize the whole manuscript.

We have substantially rewritten the manuscript. This process included standardising the use of language, combining the results and discussion section into a more cohesive whole and reordering sections for improved flow.

**From annotated pdf**

(Figure 13) You should take into account colorblind people (so red and green so close are not the best choice).

Green color replaced with black in this figure.

**Itemised Tracked changes**

The tracked changes pdf produced from latexdiff shows the changes made above to the original manuscript. In addition to changes in the text, figures 1, 2, 3, 4, 5, 6, 10, 11, 12, 13  have been improved following the reviewers recommendations. The figure order has been changed so that figures are referenced in the correct order.